# LDDMM meets GANs: Generative Adversarial Networks for diffeomorphic registration

Ubaldo Ramon, Monica Hernandez, and Elvira Mayordomo with the ADNI Consortium

Department of Computer Science and Systems Engineering, School of Engineering and Architecture, University of Zaragoza, Spain
{uramon,mhg,elvira}@unizar.es

**Abstract.** The purpose of this work is to contribute to the state of the art of deep-learning methods for diffeomorphic registration. We propose an adversarial learning LDDMM method for pairs of 3D mono-modal images based on Generative Adversarial Networks. The method is inspired by the recent literature on deformable image registration with adversarial learning. We combine the best performing generative, discriminative, and adversarial ingredients from the state of the art within the LDDMM paradigm. We have successfully implemented two models with the stationary and the EPDiff-constrained non-stationary parameterizations of diffeomorphisms. Our unsupervised learning approach has shown competitive performance with respect to benchmark supervised learning and model-based methods.

**Keywords:** Large Deformation Diffeomorphic Metric Mapping , Generative Adversarial Networks, geodesic shooting, stationary velocity fields.

## 1  Introduction

Since the 80s, deformable image registration has become a fundamental problem in medical image analysis [1]. A vast literature on deformable image registration methods exists, providing solutions to important clinical problems and applications. Up to the ubiquitous success of methods based on Convolutional Neural Networks (CNNs) in computer vision and medical image analysis, the great majority of deformable image registration methods were based on energy minimization models [2]. This traditional approach is model-based or optimization-based, in contrast with recent deep-learning approaches that are known as learning-based or data-based. Diffeomorphic registration constitutes the inception point in Computational Anatomy studies for modeling and understanding population trends and longitudinal variations, and for establishing relationships between imaging phenotypes and genotypes in Imaging Genetics [3, 4]. Model-based diffeomorphic image registration is computationally costly. In fact, the huge computational complexity of large deformation diffeomorphic metric mapping (LDDMM) [5] is considered the curse of diffeomorphic registration, where very original solutions such as the stationary parameterization [6–8], the EPDiff constraint on the initial velocity field [9], or the band-limited parameterization [10]

have been proposed to alleviate the problem. Since the advances that made it possible to learn the optical flow using CNNs (FlowNet [11]), dozens of deep-learning data-based methods have been proposed to approach the problem of deformable image registration in different clinical applications [12], some specifically for diffeomorphic registration [13–22]. Overall, all data-based methods yield fast inference algorithms for diffeomorphism computation once the difficulties with training have been overcome. Generative Adversarial Networks (GANs) is an interesting unsupervised approach where some interesting proposals for non-diffeomorphic deformable registration have been made [23] (2D) and [24, 25] (3D). GANs have also been used for diffeomorphic deformable template generation [26], where the registration sub-network is based on an established U-net architecture [22, 27], or for finding deformations for other purposes like interpretation of disease evidence [28]. A GAN combines the interaction of two different networks during training: a generative network and a discrimination network. The generative network itself can be regarded as an unsupervised method that, once included in the GAN system, is trained with the feedback of the discrimination network. The discriminator helps further update the generator during training with information regarding how the appearance of plausible warped source images. The main contribution of this work is the proposal of a GAN-based unsupervised learning LDDMM method for pairs of 3D mono-modal images, the first to use GANs for diffeomorphic registration. The method is inspired by the recent literature for deformable image registration with adversarial learning [24, 25] and combines the best performing components within the LDDMM paradigm. We have successfully implemented two models for the stationary and the EPDiff-constrained non-stationary parameterizations and demonstrate the effectiveness of our models in both 2D simulated and 3D real brain MRI data.

## 2    Background on LDDMM

Let $\Omega \subseteq \mathbb{R}^d$ be the image domain. Let $Diff(\Omega)$ be the LDDMM Riemannian manifold of diffeomorphisms and $V$ the tangent space at the identity element. $Diff(\Omega)$ is a Lie group, and $V$ is the corresponding Lie algebra [5]. The Riemannian metric of $Diff(\Omega)$ is defined from the scalar product in $V$, $\langle v, w \rangle_V = \langle Lv, w \rangle_{L^2}$, where $L$ is the invertible self-adjoint differential operator associated with the differential structure of $Diff(\Omega)$. In traditional LDDMM methods, $L = (Id - \alpha\Delta)^s, \alpha > 0, s \in \mathbb{R}$ [5]. We will denote with $K$ the inverse of operator $L$. Let $I_0$ and $I_1$ be the source and the target images. LDDMM is formulated from the minimization of the variational problem

$$E(v) = \frac{1}{2} \int_0^1 \langle Lv_t, v_t \rangle_{L^2} dt + \frac{1}{\sigma^2} \|I_0 \circ (\phi_1^v)^{-1} - I_1\|_{L^2}^2. \tag{1}$$

The LDDMM variational problem was originally posed in the space of time-varying smooth flows of velocity fields, $v \in L^2([0,1], V)$. Given the smooth flow $v : [0,1] \to V$, $v_t : \Omega \to \mathbb{R}^d$, the solution at time $t = 1$ to the evolution equation

$$\partial_t(\phi_t^v)^{-1} = -v_t \circ (\phi_t^v)^{-1} \tag{2}$$

with initial condition $(\phi_0^v)^{-1} = id$ is a diffeomorphism, $(\phi_1^v)^{-1} \in Diff(\Omega)$. The transformation $(\phi_1^v)^{-1}$, computed from the minimum of $E(v)$, is the diffeomorphism that solves the LDDMM registration problem between $I_0$ and $I_1$. The most significant limitation of LDDMM is its large computational complexity. In order to circumvent this problem, the original LDDMM variational problem is parameterized on the space of initial velocity fields

$$E(v_0) = \frac{1}{2}\langle Lv_0, v_0 \rangle_{L^2} + \frac{1}{\sigma^2}\|I_0 \circ (\phi_1^v)^{-1} - I_1\|_{L^2}^2. \tag{3}$$

where the time-varying flow of velocity fields $v$ is obtained from the EPDiff equation

$$\partial_t v_t + K[(Dv_t)^T \cdot Lv_t + DLv_t \cdot v_t + Lv_t \cdot \nabla \cdot v_t] = 0 \tag{4}$$

with initial condition $v_0$ (geodesic shooting). The diffeomorphism $(\phi_1^v)^{-1}$, computed from the minimum of $E(v_0)$ via Equations 4 and 2, verifies the momentum conservation constraint (MCC) [29], and, therefore, it belongs to a geodesic path on $Diff(\Omega)$. Simultaneously to the MCC parameterization, a family of methods was proposed to further circumvent the large computational complexity of the original LDDMM [6–8]. In all these methods, the time-varying flow of velocity fields $v$ is restricted to be steady or stationary [30]. In this case, the solution does not belong to a geodesic.

## 3 Generative Adversarial Networks for LDDMM

Similarly to model-driven approaches for estimating LDDMM diffeomorphic registration, data-driven approaches for learning LDDMM diffeomorphic registration aim at the inference of a diffeomorphism $(\phi_1^v)^{-1}$ such that the LDDMM energy is minimized for a given $(I_0, I_1)$ pair. In particular, data-driven approaches compute an approximation of the functional

$$\mathcal{S}(\arg\min_{v \in V} E(v, I_0, I_1)) \tag{5}$$

where $\mathcal{S}$ represents the operations needed to compute $(\phi_1^v)^{-1}$ from $v$, and the energy $E$ is either given by Equations 1 or 3. The functional approximation is obtained via a neural network representation with parameters learned from a representative sample of image pairs. Unsupervised approaches assume that the LDDMM parameterization in combination with the minimization of the energy $E$ considered as a loss function are enough for the inference of suitable diffeomorphic transformations after training. Therefore, there is no need for ground truth deformations. GAN-based approaches depart from unsupervised approaches by the definition of two different networks: the generative network (G) and the discrimination network (D), and are trained in an adversarial fashion as follows. The discrimination network D learns to distinguish between a warped source image $I_0 \circ (\phi_1^v)^{-1}$ generated by G and a plausible warped source image. It is trained using the loss function

$$L_D = \begin{cases} -\log(p) & c \in P^+ \\ -\log(1-p) & c \in P^- \end{cases} \tag{6}$$

where $c$ indicates the input case, $P^+$ and $P^-$ indicate positive or negative cases for the GAN, and $p$ is the probability computed by D for the input case. In the first place, D is trained on a positive case $c \in P^+$ representing a target image $I_1$ and a warped source image $I_0^w$ plausibly registered to $I_1$ with a diffeomorphic transformation. The warped source image is modeled from $I_0$ and $I_1$ with a strictly convex linear combination: $I_0^w = \beta I_0 + (1 - \beta)I_1$. It should be noticed that, although the warped source image would ideally be $I_1$, the selection of $I_0^w = I_1$ (e.g. $\beta = 0$) empirically leads to the discriminator rapidly outperforming the generator. This approach to discriminators has been successfully used in adversarial learning methods for deformable registration [25]. Next, D is trained on a negative case $c \in P^-$ representing a target image $I_1$ and a warped source image $I_0^w$ obtained from the generator network G. The generative network in this context is the diffeomorphic registration network. G is aimed at the approximation of the functional given in Equation 5 similarly to unsupervised approaches for the inference of $(\phi_1^v)^{-1}$. It is trained using the combined loss function

$$L_G = L_{\text{adv}} + \lambda E(v, I_0, I_1). \tag{7}$$

where $L_{\text{adv}}$ is the adversarial loss function, defined from $L_{\text{adv}} = -\log(p)$ where $p$ is computed from D; $E$ is the LDDMM energy given by Equations 1 or 3; and $\lambda$ is the weight for balancing the adversarial and the generative losses. For each sample pair $(I_0^w, I_1)$, G is fed with the pair of images and updates the network parameters from the back-propagation of the information of the loss function values coming from the LDDMM energy and the discriminator probability of being a pair generated by G.

### 3.1   Proposed GAN architecture

**Generator network.** In this work, the diffeomorphic registration network G is intended to learn LDDMM diffeomorphic registration parameterized on the space of steady velocity fields or the space of initial velocity fields subject to the EPDiff equation (Equation 4). The diffeomorphic transformation $(\phi_1^v)^{-1}$ is obtained from these velocity fields either from scaling and squaring [7, 8] or the solution of the deformation state equation [5]. Euler integration is used as PDE solver for all the involved differential equations. A number of different generator network architectures have been proposed in the recent literature, with predominance of simple fully convolutional (FC) [23] or U-Net like architectures [24, 25]. In this work, we propose to use the architecture by Duan et al. [24] adapted to fit our purposes. The network follows the general U-net design of utilizing an encoder-decoder structure with skip connections. However, during the encoding phase, the source and target images are fed to two encoding streams with different resolution levels. The combination of the two encoding streams allows a larger receptive field suitable to learn large deformations. The upsampling is performed with a deconvolutional operation based on transposed convolutional layers [31]. We have empirically noticed that the learnable parameters of these layers help reduce typical checkerboard GAN artifacts in the decoding [32].

**Discriminator network.** The discriminator network D follows a traditional CNN architecture. The two input images are concatenated and passed through five convolutional blocks. Each block includes a convolutional layer, a RELU activation function, and a size-two max-pooling layer. After the convolutions, the 4D volume is flattened and passed through three fully connected layers. The output of the last layer is the probability of the input images to come from a registered pair not generated by G.

**Generative-Discriminative integration layer.** The generator and the discriminator networks G and D are connected through an integration layer. This integration layer allows calculating the diffeomorphism $(\phi_1^v)^{-1}$ that warps the source image $I_0$. The selected integration layer depends on the velocity parameterization: stationary (SVF-GAN) or EPDiff-constrained time-dependent (EPDiff-GAN). The computed diffeomorphisms are applied to the source image via a second 3D spatial transformation layer [33] with no learnable parameters.

**Parameter selection and implementation details** We selected the parameters $\lambda = 1000$, $\sigma^2 = 1.0$, $\alpha = 0.0025$, and $s = 4$ and a unit-domain discretization of the image domain $\Omega$ [5]. Scaling and squaring and Euler integration were performed in 8 and 10 time samples respectively. The parameter $\beta$ for the convex linear modeling of warped images was selected equal to 0.2. Both the generator network and the discriminator network were trained with Adam's optimizer with default parameters and learning rates of $5e^{-5}$ for G and $1e^{-6}$ for D, respectively. The experiments were run on a machine equipped with one NVidia Titan RTX with 24 GBS of video memory and an Intel Core i7 with 64 GBS of DDR3 RAM, and developed in Python with Keras and a TensorFlow backend.

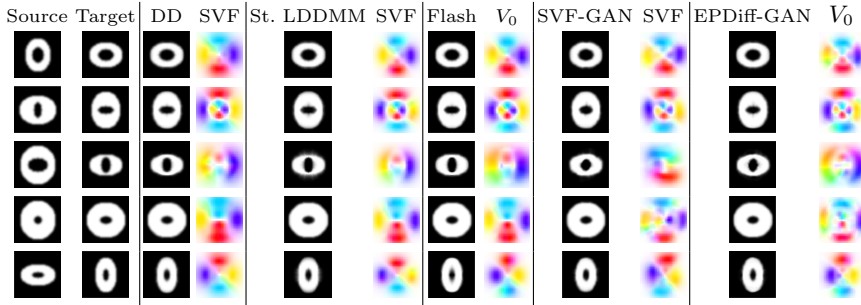

**Fig. 1.** Example of simulated 2D registration results. Up: source and target images of five selected experiments. Down, left to right: deformed images and velocity fields computed from diffeomorphic Demons (DD), stationary LDDMM (St. LDDMM), Flash, and our proposed SVF-GAN and EPDiff-GAN. SVF stands for a stationary velocity field and $V_0$ for the initial velocity field of a geodesic shooting approach, respectively.

## 4    Experiments and Results

**2D simulated dataset.** We simulated a total of 2560 torus images by varying the parameters of two ellipse equations, similarly to [19]. The parameters were drawn from two Gaussian distributions: $\mathcal{N}(4, 2)$ for the inner ellipse and $\mathcal{N}(12, 4)$ for the outer ellipse. The simulated images were of size $64 \times 64$. The networks were trained during 1000 epochs with a batch size of 64 samples.

**3D brain MRI datasets.** We used a total of 2113 T1-weighted brain MRI images from the Alzheimer's Disease Neuroimaging Initiative (ADNI). The images were acquired at the baseline visit and belong to all the available ADNI projects (1, 2, Go, and 3). The images were preprocessed with N3 bias field correction, affinely registered to the MNI152 atlas, skull-stripped, and affinely registered to the skull-stripped MNI152 atlas. The evaluation of our generated GAN models in the task of diffeomorphic registration was performed in NIREP dataset [34], where one image was chosen as reference and pair-wise registration was performed with the remaining 15. All images were scaled to size $176 \times 224 \times 176$, and in this case trained for 50 epochs with a batch size of 1 sample. Inference of either a stationary or a time dependent velocity field takes 1.3 seconds.

**Results in the 2D simulated dataset** Figure 1 show the deformed images and the velocity fields obtained in the 2D simulated dataset by diffeomorphic Demons [7], a stationary version of LDDMM (St. LDDMM) [8], the spatial version of Flash [10], and our proposed SVF and EPDiff GANs. Apart from diffeomorphic Demons that uses Gaussian smoothing for regularization, all the considered methods use the same parameters for operator $L$. Therefore, St. LDDMM and SVF-GAN can be seen as a model-based and a data-based approach for the minimization of the same variational problem. The same happens with Flash and EPDiff-GAN. From the figure, it can be appreciated that our proposed GANs are able to obtain accurate warps of the source to the target images, similarly to model-based approaches. For SVF-GAN, the inferred velocity fields are visually similar to model-based approaches in three of five experiments. For EPDiff-GAN, the inferred initial velocity fields are visually similar to model-based approaches in four of five experiments.

### 4.1    Results in the 3D NIREP dataset

**Quantitative assessment** Figure 2 shows the Dice similarity coefficients obtained with diffeomorphic Demons [7], St. LDDMM [8], Voxelmorph II [16], the spatial version of Flash [10], Quicksilver [14] and our proposed SVF and EPDiff GANs. SVF-GAN shows an accuracy similar to St. LDDMM and competitive with diffeomorphic Demons. Our proposed method tends to overpass Voxelmorph II in the great majority of the structures. On the other hand, EPDiff-GAN shows an accuracy similar to Flash and Quicksilver in the great majority of regions, with the exception of the temporal pole (TP) and the orbital frontal gyrus

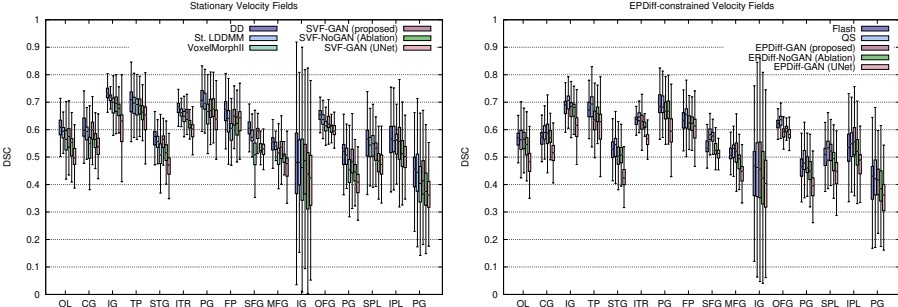

**Fig. 2.** Evaluation in NIREP. Dice scores obtained by propagating the diffeomorphisms to the segmentation labels on the 16 NIREP brain structures. Left, methods parameterized with stationary velocity fields: diffeomorphic Demons (DD), stationary LDDMM (St. LDDMM), Voxelmorph II, our proposed SVF-GAN with the two-stream architecture, SVF-GAN without discriminator and SVF-GAN with a U-net. Right, geodesic shooting methods: Flash, Quicksilver (QS), our proposed EPDiff-GAN, EPDiff-GAN without discriminator, and EPDiff-GAN with a U-net.

(OFG), two small localized and difficult to register regions. Furthermore, the two-stream architecture greatly improves the accuracy obtained by a simple U-Net. SVF-GAN outperforms the ablation study model in which no discriminator was used, though EPDiff-GAN only shows clear performance improvements in some structures. It drives our attention that Flash underperformed in the superior frontal gyrus (SFG). All tested methods generate smooth deformations with almost no foldings, as can be seen in table 1 from the supplementary material.

**Qualitative assessment** For a qualitative assessment of the quality of the registration results, Figure 3 shows the sagittal and axial views of one selected NIREP registration result. In the figure, it can be appreciated a high matching between the target and the warped ventricles, and more difficult to register regions like the cingulate gyrus (observable in the sagittal view) or the insular cortex (observable in the axial view).

## 5    Conclusions

We have proposed an adversarial learning LDDMM method for the registration of 3D mono-modal images. We have successfully implemented two models: one for the stationary parameterization and the other for the EPDiff-constrained non-stationary parameterization (geodesic shooting). The performed ablation study shows how GANs improve the results of the proposed registration networks. Furthermore, our experiments have shown that the inferred velocity fields are comparable to the solutions of model-based approaches. In addition, the evaluation study has shown the competitiveness of our approach with state of the

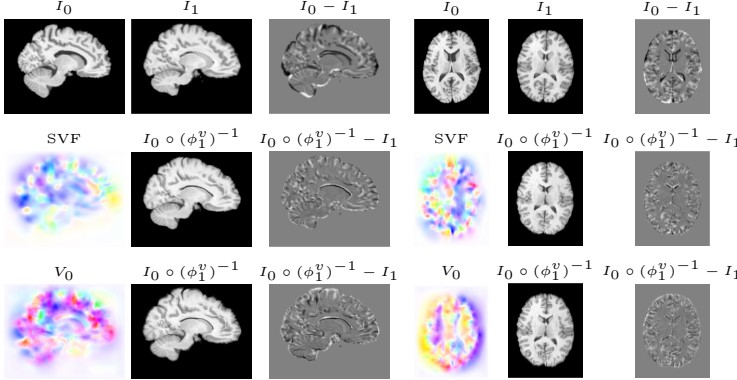

**Fig. 3.** Example of 3D registration results. First row, sagittal and axial views of the source and the target images and the differences before registration. Second row, inferred stationary velocity field, warped image, and differences after registration for SVF-GAN. Third row, inferred initial velocity field, warped image, and differences after registration for EPDiff-GAN.

art model- and data- based methods. It should be remarked that our methods perform similarly to Quicksilver, a supervised method that uses patches for training, and therefore, it learns in a rich-data environment. In contrast, our method is unsupervised and uses the whole image for training in a data-hungry environment. Indeed, our proposed methods outperform Voxelmorph II, an unsupervised method for diffeomorphic registration usually selected as benchmark in the state of the art. Finally, our proposal may constitute a good candidate for the massive computation of diffeomorphisms in Computational Anatomy studies, since once training has been completed, our method shows a computational time of over a second for the inference of velocity fields.

**Acknowledgements** This work was partially supported by the national research grant TIN2016-80347-R (DIAMOND project), PID2019-104358RB-I00 (DL-Ageing project), and Government of Aragon Group Reference $T64\_20R$ (COSMOS research group). Ubaldo Ramon-Julvez's work was partially supported by an Aragon Government grant. Project PID2019-104358RB-I00 granted by MCIN/AEI/10.13039/501100011033. We would like to thank Gary Christensen for providing the access to the NIREP database [34]. Data used in the preparation of this article were partially obtained from the Alzheimer's Disease Neuroimaging Initiative (ADNI) database (adni.loni.usc.edu). As such, the investigators within the ADNI contributed to the design and implementation of ADNI and/or provided data but did not participate in the analysis or writing of this report. A complete listing of ADNI investigators can be found at: `https://adni.loni.usc.edu/wp-content/uploads/how_to_apply/ADNI_Acknowledgement_List.pdf`.

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
