# Supplementary material of LDDMM meets GANs: Generative Adversarial Networks for diffeomorphic registration

Ubaldo Ramon, Monica Hernandez, Elvira Mayordomo, and with the ADNI Consortium

Department of Computer Science and Systems Engineering, School of Engineering and Architecture,
University of Zaragoza, Spain
{uramon,mhg,elvira}@unizar.es

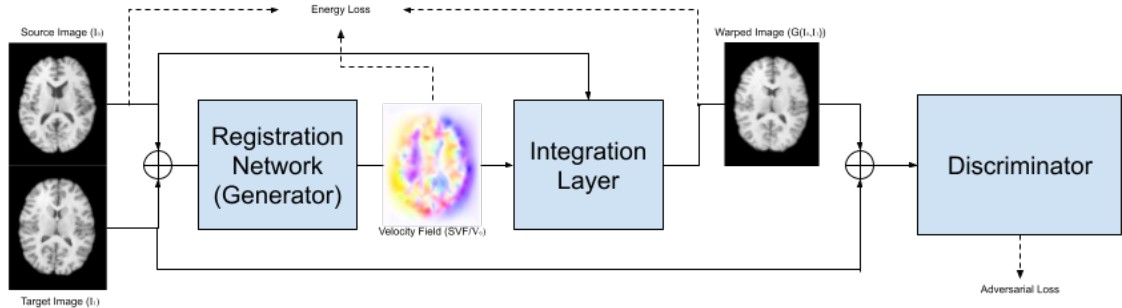

**Fig. 1.** Overview of the whole GAN diffeomorphic registration architecture. The Generator takes as input the concatenated source and target images, and outputs a stationary or time dependant velocity field. The velocity field is then feed to the integration layer (which depends on the diffeomorphism parameterization) and outputs the final transformation with which the warped image is calculated. Finally, the Discriminator is given the concatenated warped and target images, and the Energy loss is calculated.

| Model | $MSE_{rel}$ | DSC | % of $|J_{\phi^{-1}}| < 0$ | % of $|J_{\phi^{-1}}| > 10$ |
|---|---|---|---|---|
| QS | $0.190 \pm 0.013$ | $0.575 \pm 0.097$ | $< 0.001$ | $< 0.001$ |
| VM-II (Disp) | $0.144 \pm 0.010$ | $0.583 \pm 0.102$ | $1.16 \pm 0.13$ | $0.04 \pm 0.001$ |
| VM-II (SVF) | $0.227 \pm 0.009$ | $0.555 \pm 0.102$ | $< 0.001$ | $0.01 \pm 0.003$ |
| SVF-GAN (proposed) | $0.224 \pm 0.015$ | $0.576 \pm 0.096$ | $< 0.001$ | $< 0.001$ |
| EPDiff-GAN | $0.235 \pm 0.017$ | $0.557 \pm 0.097$ | $< 0.001$ | $< 0.001$ |
| SVF-Ablation | $0.272 \pm 0.021$ | $0.545 \pm 0.101$ | $< 0.001$ | $< 0.001$ |
| EPDiff-Ablation | $0.265 \pm 0.021$ | $0.545 \pm 0.100$ | $< 0.001$ | $< 0.001$ |
| SVF-Unet | $0.307 \pm 0.011$ | $0.521 \pm 0.103$ | $< 0.001$ | $< 0.001$ |
| EPDiff-Unet | $0.394 \pm 0.018$ | $0.498 \pm 0.100$ | $< 0.001$ | $< 0.001$ |

**Table 1.** Evaluation in NIREP of benchmark deeplearning methods, proposed methods and ablation tests. All measures show the mean and standard deviation across the 15 registrations. From left to right, : Deep learning model used for registration, relative mean squared error, DICE score, percentage of negative jacobian determinants, percentage of jacobian determinants higher than 10. Deep learning methods, from top to bottom: Quicksilver (QS), Voxelmorph II both displacement (DISP) and stationary velocity field parameterization versions (SVF) , proposed SVF-GAN, proposed EPDiff-GAN, ablation test SVF without GAN, ablation test EPDiff without GAN, SVF-GAN with Unet registration architecture and EPDiff-GAN with Unet registration architecture.

**Acknowledgements** Data collection and sharing for this project was funded by the Alzheimer's Disease Neuroimaging Initiative (ADNI) (National Institutes of Health Grant U01 AG024904) and DOD ADNI (Department of Defense award number W81XWH-12-2-0012). ADNI is funded by the National Institute on Aging, the National Institute of Biomedical Imaging and Bioengineering, and through generous contributions from the following: AbbVie, Alzheimer's Association; Alzheimer's Drug Discovery Foundation; Araclon Biotech; BioClinica, Inc.; Biogen; Bristol-Myers Squibb Company; CereSpir, Inc.; Cogstate; Eisai Inc.; Elan Pharmaceuticals, Inc.; Eli Lilly and Company; EuroImmun; F. Hoffmann-La Roche Ltd and its affiliated company Genentech, Inc.; Fujirebio; GE Healthcare; IXICO Ltd.; Janssen Alzheimer Immunotherapy Research & Development, LLC.; Johnson & Johnson Pharmaceutical Research & Development LLC.; Lumosity; Lundbeck; Merck & Co., Inc.; Meso Scale Diagnostics, LLC.; NeuroRx Research; Neurotrack Technologies; Novartis Pharmaceuticals Corporation; Pfizer Inc.; Piramal Imaging; Servier; Takeda Pharmaceutical Company; and Transition Therapeutics. The Canadian Institutes of Health Research is providing funds to support ADNI clinical sites in Canada. Private sector contributions are facilitated by the Foundation for the National Institutes of Health (www.fnih.org).

The grantee organization is the Northern California Institute for Research and Education, and the study is coordinated by the Alzheimer's Therapeutic Research Institute at the University of Southern California. ADNI data are disseminated by the Laboratory for Neuro Imaging at the University of Southern California.