# OpenReview forum: "LDDMM meets GANs: Generative Adversarial Networks for diffeomorphic registration"
_WBIR.info/2022/Workshop/Biomedical_Imaging_Registration — WBIR 2022_

### Official Review · Reviewer_phm5 · 2022-02-12

**Rating:** 4
**Confidence:** 5

**Deanonymize Review:**

no

**Detailed Comments:**

Comment:

Demonstrating how the adversarial loss improves/affects the registration will be beneficial to this work. It is also interesting to see the effect of the hyperparameter \lambda. Could the method be trained without the similarity/distance measure (in equation 3)?

Evaluating the diffeomorphic properties and smoothness of the proposed method by measuring the Jacobian determinant of the solutions would further improve this paper.

Adding quantitative results of the 2D simulated dataset will help readers better differentiate the registration performance of different LDDMM methods.

Minor:

Typo: is considered “as” the curse …

Typo: V “be” the tangent space …

Typo: On page 4, “It” is trained using the loss function …

------------------------------------

Overall, the combination and evaluation of GAN and two different diffeomorphic transformation models are well-grounded efforts. This paper makes a useful contribution to the learning-based LDDMM literature. But this paper is held-back by the unclear presentation of experiments and insufficient evaluation of deformation field/GAN loss. The content and scope of this paper are of interest to the WBIR community.


**Paper Type:**

both

**Strengths Weaknesses:**

Summary:
This paper proposes a learning-based large deformation diffeomorphic metric mapping (LDDMM) method with generative adversarial networks (GANs). The generator aims to estimate the diffeomorphic transformation that aligns the source and target images, where the discriminator aims to discriminate the generator’ solutions and plausibly registered solutions, and guide the generator to output plausible solutions. The method is demonstrated with stationary velocity fields (SVF) and initial velocity fields subject to the EPDiff equation. The method is evaluated on 2D simulated and 3D brain MRI datasets (NIREP). Experiments show that the proposed method achieves comparable registration accuracy with two conventional LDDMM methods and three learning-based methods.
------------------------------------

Strengths:

Although neither the adversarial learning nor LDDMM formulation is new, the combination and evaluation of GAN and two different diffeomorphic transformation models are well-grounded efforts.

The paper is well-written and easy to follow.

5 LDDMM/learning-based methods are compared with.

The literature and background on LDDMM methods are sufficient and clear. Interested readers are able to understand the methodology without domain knowledge in LDDMM.

Slight improvement in 3D brain MRI registration with the proposed two-stream architecture.

------------------------------------
Weaknesses:

The improvement and impact of adversarial learning are unclear. It is clear that the generative network can be trained without the adversarial loss (in equation 7). An ablation study showing the effect of the adversarial loss is necessary as the main focus of this work is a proper combination of GANs and LDDMM.

Lack of evaluations on the smoothness and diffeomorphic properties of the resulting deformation field. In deformable registration, there is a common trade-off between registration accuracy and smoothness of the deformation field. Also, although the diffeomorphic properties are theoretically guaranteed by the transformation model, there are small violations of the diffeomorphic properties, e.g., 1-1 mapping/invertibility, due to limited (discrete) time steps and interpolation in practice.

The improvement over the existing methods is not significant. In most of the anatomical structures, the registration performance of the proposed method is slightly inferior to existing LDDMM methods (DD, St. LDDMM).

The experiment setting of 3D brain registration is unclear. How was the reference image(s) selected in the experiment? Is it a pairwise registration task or an atlas-based registration task? How many pairs of testing data were involved in the testing phase?

---

### Official Review · Reviewer_GGt2 · 2022-02-16

**Rating:** 4
**Confidence:** 3
**Recommendation:** Long Oral, Short Oral

**Deanonymize Review:**

no

**Detailed Comments:**

__Remarks:__

Introduction:
* I really liked the well-structured introduction which gives a nice overview about the existing work
*  other LDDMM methods using GANs are mentioned, but a differentiation from these is missing. Please add further details on what is special/new about the proposed method.

Generative Adversarial Networks for LDDMM:
* Overall comprehensible, but adding a diagram/illustration how discriminator, generator and LDDMM play together could be helpful
* 2 typing errors/remarks:
    * _...a plausible warped source image. **Is** is trained using the loss function..._
    *	_The warped source image is modeled from a strictly convex linear combination of I_0 and I_1_  please rephrase or add something after I_1 so that the equation does not start directly after I_1, otherwise one gets confused that I_1 also belongs to the equation

Proposed GAN architecture:
* A clear statement of what is new/different from other approaches that use GANs should be added. Especially in the subsection _Generator network_ it is not clear, if the explanation of the 2 encoding streams belongs to the mentioned architecture [24] or to the adaptions made.

Results:
* What is the reason for selecting these two datasets? Consider including a statement in the paper explaining the purpose.
* As the proposed architecture is derived from [24], I expected a comparison of the results with this or other LDDMM-GAN approaches. Please add that, if it is possible.


__Overall:__

I suggest to accept this paper as it is a well-written, solid paper and and the presented method is interesting.  However, the novelty is not clearly stated, which is why I only weakly accept the work.

**Paper Type:**

validation / application paper

**Strengths Weaknesses:**

This paper presents a learning method for solving the LDDMM image registration problem using a GAN-based approach. Thereby, the generator of the GAN is supposed to find solutions the registration problem, while the discriminator of the GAN tries to distinguish the generated solutions from plausible solutions. The presented method has been tested with synthetic and real data and is comparable to state-of-the-art methods.

__Strenghts__:
- well-written and well-structured paper, compact but comprehensible introduction to LDDMM and GANs
- good overview of prior work and state-of-the-art
- solid experiments and comparison with other methods

__Weaknesses__:
- unclear novelty, adaption of the architecture of an existing approach, comparison of results with that approach missing
- goal of the paper not clearly recognisable

---

### Official Review · Reviewer_o5At · 2022-02-20

**Rating:** 4
**Confidence:** 3
**Recommendation:** Short Oral

**Deanonymize Review:**

no

**Detailed Comments:**

- Figure 2 would be easier to overview if it were larger
- description of the implementation of the model architecture could be more detailed
- publicly available source code would be great

The presented method does not yield outstanding results and a more detailed evaluation would be desirable. Nevertheless, the idea of the GAN-based approach for diffeomorphic image registration is interesting and a revised version of this paper is worth being presented at WBIR 2022.

**Paper Type:**

methodological development

**Strengths Weaknesses:**

(+) GAN-based approach for diffeomorphic image registration

(+) two different proposed models: SVF-GAN and EPDiff-GAN

(+) Dice scores on NIREP dataset on par with chosen comparison methods

(-) no quantitative 2D results given

(-) qualitative 3D results (warped images) show some distortions which result in the fact that the anatomy does not appear completely plausible

The paper describes a new GAN-based method vor diffeomorphic image registration. The authors propose two different models for stationary and non-stationary parameterisation that both lead to Dice scores on par with the chosen comparison methods when used for registration of the NIREP dataset. The qualitative (3D) registration results show some distortions that make the warped images appearing not completely plausible. Maybe this could be addressed by regularisation of the velocity fields or a higher weighting of L_adv. The quantitative results could generally be more detailed. For 3D, only Dice scores for one dataset are given and for 2D no quantitative results are reported. If reporting results for 2D, it would be interesting how the method performs on non-simulated data, e.g. middle slices of brain MRI.
Overall, structure and language of the paper are appropriate, although a clearer separation of the methodological descriptions of the two models would be beneficial.

---

### Official Review · Reviewer_fBe2 · 2022-02-20

**Rating:** 4
**Confidence:** 4
**Recommendation:** Long Oral

**Deanonymize Review:**

no

**Detailed Comments:**

Minor: above eq 6: “Is is”. Several spelling and grammar mistakes “empiricaly”. A full proof read is advised.
The detail of the timing for training the model is unnecessary.
Another contemporary piece of work to discuss is [1], which uses GANs, but without diffeomorphic transformations.

Is \beta fixed over time? It feels like it should be randomly sampled, drawing some comparison with the Wasserstein GAN approach.

In Fig 3: are V0 and the SVF similarly interpretable? It would be nice to have the displacement field available (possibly in the supplement) for comparison. V_0 seems much smoother than the SVF, and it would be useful to provide a direct comparison of this.

Some form of network diagram would be helpful in describing your two stream approach - possibly in the supplementary material.

[1] Bigolin Lanfredi, Ricardo, et al. "Interpretation of disease evidence for medical images using adversarial deformation fields." International Conference on Medical Image Computing and Computer-Assisted Intervention. Springer, Cham, 2020.

Summary:
This work proposes an interesting combination of diffeomorphic registration and GANs.  Although the theoretical justification is a little limited, I think this could be expanded on to provide an informative talk at WBIR.

**Paper Type:**

methodological development

**Strengths Weaknesses:**

Strengths:
This paper proposes the integration of GANs with LDDMM transformation models. The LDDMM formulation (and EPDiff) solution is clearly explained, and the integration with the GAN loss seems reasonable. Some investigation is carried out into model architectures, and a two-stream approach was found to perform better than a U-net.

Weaknesses:
This paper would be substantially improved by providing an argument why you  might want to combine these things, and what benefits you might expect from the GAN based training. The level of discussion to justify this choice is currently too limited, and only weakly supported by the experimental data. I’m also not sure that we learn much from the synthetic experiments  - there needs to be some more discussion of this.

---

### Decision · Program_Chairs · 2022-02-22

Accept